# LiDAR Technology for UAV Detection: From Fundamentals and Operational Principles to Advanced Detection and Classification Techniques

**DOI:** 10.3390/s25092757

**Published:** 2025-04-27

**Authors:** Ulzhalgas Seidaliyeva, Lyazzat Ilipbayeva, Dana Utebayeva, Nurzhigit Smailov, Eric T. Matson, Yerlan Tashtay, Mukhit Turumbetov, Akezhan Sabibolda

**Affiliations:** 1Metropolitan College (MET), Boston University, Boston, MA 02215, USA; useidali@bu.edu; 2Department of Radio Engineering, Electronics and Telecommunications, International IT University, Almaty 050040, Kazakhstan; 3Department of Computer Science, SDU University, Almaty 040900, Kazakhstan; 4Department of Electronics, Telecommunications and Space Technologies, Satbayev University, Almaty 050013, Kazakhstanm.turumbetov@satbayev.university (M.T.); a.sabibolda@satbayev.university (A.S.); 5Department of Mechanical Engineering and Robotics, U. A. Joldasbekov Institute of Mechanics and Engineering, Almaty 050010, Kazakhstan; 6Department of Computer and Information Technology, Purdue University, West Lafayette, IN 47907-2021, USA; ematson@purdue.edu

**Keywords:** drone detection, unmanned aerial vehicles (UAVs), UAV detection, object detection, LiDAR, LiDAR classifications, scanning mechanism, deep learning, point clouds, deep learning for point cloud processing, 3D object detection

## Abstract

As unmanned aerial vehicles (UAVs) are increasingly employed across various industries, the demand for robust and accurate detection has become crucial. Light detection and ranging (LiDAR) has developed as a vital sensor technology due to its ability to provide rich 3D spatial information, particularly in applications such as security and airspace monitoring. This review systematically explores recent innovations in LiDAR-based drone detection, deeply focusing on the principles and components of LiDAR sensors, their classifications based on different parameters and scanning mechanisms, and the approaches for processing LiDAR data. The review briefly compares recent research works in LiDAR-based only and its fusion with other sensor modalities, the real-world applications of LiDAR with deep learning, as well as the major challenges in sensor fusion-based UAV detection.

## 1. Introduction

The rapid development and increasing use of unmanned aerial vehicles (UAVs) in a variety of industries, from agriculture and medicine to environmental monitoring and entertainment, has created both new opportunities and concerns, notably in terms of airspace control and security. Since drones are capable of transporting illegal and suspicious cargo ranging from drugs to dangerous explosives, drone incidents have become more frequent in recent times. For example, in the first quarter of 2025, the FAA (Federal Aviation Administration) reported more than 400 drone-related near-miss incidents near U.S. airports, with multiple cases requiring pilots to take evasive action [1]. Drones were detected to bring drugs, mobile phones, and other illicit things to correctional institutions in both Canada and the UK [2]. In addition, in May 2023, Mexican criminal organizations employed drones to hurl homemade explosives into villages along drug trafficking routes [3].

Significance of the research work. Several drone incidents cited above demonstrate that timely detection of a suspicious UAV is crucial for a number of reasons spanning a variety of industries and applications:In the field of security and defense, the ability of timely UAV detection prevents dangerous activities, such as espionage, smuggling, and terrorist attacks;Drone identification on the issue of confidentiality allows for the protection of individuals and organizations from unauthorized control;Timely detection of suspicious UAVs in the matter of maintaining the safety of the airways allows us to prevent an air collision.

However, there are numerous challenges in detecting suspicious UAVs. In particular, the variety of payloads, different operating environments, the small size of the UAV, its continuous dynamic movement, and the continuous development of drone technology require complex and reliable detection methods. Currently, UAV detection, classification, and tracking algorithms may employ numerous sensors, such as radars, RF (radio frequency) sensors, acoustic sensors, cameras, or a fusion of them (see Table 1).

Radar-based UAV detection [4,5,6] is essential due to improved detection accuracy, efficiency, and real-time monitoring in response to growing concerns about UAV abuse in sensitive regions such as airports and military zones. Radar-based detection systems, particularly those based on micro-Doppler signatures, have proven to be quite successful in distinguishing UAVs from other flying objects, such as birds or small aircraft. These systems can recognize distinctive UAV flight patterns due to the micro-Doppler effect, which records the slight motion of UAV rotors. This is critical for detecting UAVs that pose potential dangers in real time [4]. Therefore, radars are becoming dominant technologies used for UAV detection because of their ability to detect targets at long distances and in various weather conditions, as well as in complicated surroundings. However, their effectiveness may be limited by the low radar cross section (RCS), the high cost, and the complexity of the deployment [4,5,6].

Recent research on identifying hostile UAVs with RF sensors [7,8,9,10] has made tremendous progress, particularly in the integration of machine learning (ML) and deep learning (DL) approaches. RF sensors are capable of monitoring communication signals and spectra between drones and their operators. In [7], the authors applied a multiscale feature extraction approach to identify UAVs by examining their RF fingerprints in real time. The system used an end-to-end DL model with residual blocks that were evaluated on various signal-to-noise ratios (SNRs) and showed excellent detection accuracy (above 97%) even in noisy environments, outperforming traditional approaches that struggle with overlapping signals. Another research [8] found that RF sensors can detect not just UAVs but also their controllers, which is essential to identifying the operator of a hostile UAV. This system combines radar and RF technology to follow unlicensed drones and their command centers, providing real-time data to security teams. Therefore, RF sensors are an effective and passive means of detecting UAVs, particularly in circumstances where non-line-of-sight detection is critical. Their ability to recognize both the UAV and its operator, along with long-range detection, makes them extremely useful in security and surveillance applications. However, their shortcomings, such as dependency on active communication signals, sensitivity to encryption and jamming, and the inability to identify fully autonomous drones that do not emit RF signals, frequently demand their integration with other sensor types (e.g., radar or optical sensors) to create a more robust UAV detection system [9,10].

The use of deep learning to recognize drone sound signatures represents a major improvement in object detection. Acoustic sensors detect unique sound signatures produced by drone engines [11,12,13,14]. A recent study [11] demonstrated that the deep neural network (DNN) could interpret multirotor UAV sounds acquired by acoustic sensors and reliably differentiate UAVs from background noise. The study compares convolutional neural network (CNN) and recurrent neural network (RNN) algorithms in a single or late-fusion voting ensemble. The results of the experiment demonstrated that the CNN-based models performed best, with an accuracy of 94.7%. In another study [12], researchers evaluated the ability of UAV sound detection at various distances, examining how ambient factors such as noise and distance impact UAV detection. The research results demonstrated that linear discriminant analysis might be effective for the detection of UAV sound at short distances, while the increased detection accuracy at medium and long distances was achieved using the YAMNet DL model. Therefore, acoustic sensors are an inexpensive and energy-efficient alternative for UAV detection, particularly in locations with limited visual line-of-sight (LoS). However, their performance and detection range are reduced by wind conditions and background noise [13,14].

The use of spatio-temporal information in conjunction with optical flow analysis is a common strategy in camera-based UAV detection research [15]. This approach improves small UAV detection by evaluating continuous image sequences to capture drone motions across frames [15]. The problem of distinguishing drones from other flying objects was considered in [16,17]. In [16], the authors conducted real-time drone detection by separating the problem into moving object detection and classification tasks. The moving object detection task was solved using traditional image processing operations and the background subtraction algorithm. The MobileNetv2 DL model handled the task of classifying the detected moving objects. Although camera sensors offer rich color and texture detection and are inexpensive and adaptable for object recognition, they have drawbacks such as sensitivity to lighting, weather, and line-of-sight conditions and lack of depth perception [17]. These problems often require the incorporation of other detection systems such as acoustic [18,19], RF [20,21], radar [22,23], as well as light detection and ranging (LiDAR) [24] sensors to improve the total reliability of UAV detection.

LiDAR [24,25,26,27,28,29,30,31,32,33,34,35,36,37,38,39,40,41,42], unlike typical camera- or radar-based systems, gives exact distance measurements and high-resolution data with rich 3D representations for distinguishing tiny, fast-moving objects such as drones from other aerial objects based on motion and size characteristics. In addition, LiDAR sensors have shown promising results for UAV detection, localization, and tracking in both near- and mid-range scenarios. Because of its ability to capture precise high-resolution 3D spatial data, its durability under harsh environmental conditions, and its great precision in recognizing and tracking fast-moving objects, LiDAR is the preferred choice for applications that require robustness and accuracy. As UAV technology advances, LiDAR’s performance benefits keep it at the forefront of UAV detection systems, particularly in key applications such as airspace security, surveillance, and autonomous navigation. However, effective LiDAR data processing for UAV detection presents distinct challenges. The irregularity, sparsity, and high dimensionality of the point-cloud data necessitate sophisticated analytical approaches capable of effectively recognizing different types of drones in diverse environments. Deep learning has shown great promise in this domain, providing advanced approaches for feature extraction, object identification, and classification. The combination of LiDAR technology and deep learning methodologies is driving drone detection innovation with enhanced system accuracy, speed, and robustness. This systematic study seeks to investigate the most recent advances in LiDAR-based object detection, with a particular emphasis on the fundamentals of LiDAR technology, its structure and principle, classification types, as well as clustering-based and deep learning algorithms for LiDAR data processing. We will look at the progress of integrating LiDAR with additional sensor modalities in complex environments to increase system resilience and accuracy.

The remainder of this paper is organized as follows: Section 2 describes the fundamentals of LiDAR technology, including its working principles, structural components, and classification into scanning and non-scanning types. Section 3 presents a brief review of deep learning-based LiDAR data processing methods such as clustering-based and deep learning-based approaches, real-world applications of LiDAR with deep learning, and the major challenges in sensor fusion-based UAV detection. The paper concludes by summarizing the main idea of the research and recommending future directions in LiDAR-based UAV detection systems.

## 2. Fundamentals of LiDAR Technology

Light Detection and Ranging (LiDAR) is a type of remote sensing technique that works similarly to radar but employs light instead of radio waves. It uses the concepts of reflected light and precise timing to determine the distance between objects [24]. However, LiDAR is more than simply a distance measurement. LiDAR systems can generate detailed 3D models of the environment by emitting laser pulses and measuring the time required for the light beam to reach the target and reflect back. This technique is highly precise and can capture complicated scenes in real time. Therefore, it may also be employed in 3D mapping and imaging, making it both desirable in an engineering setting and a very valuable practical technology.

A common LiDAR sensor utilizing a laser in the 905 nm near-infrared (NIR) spectrum employs the time-of-flight (ToF) conversion technology. ToF is considered as the time difference between the laser’s transmission (*t1*) and its reflection from a target object back to the sensor (*t2*). The idea of a single-shot direct time-of-flight (*dToF*) measurement is the simplest to understand among the various LiDAR approaches to calculate the distance to a target object [24,25,26] (see Figure 1). Based on this idea, a light source (typically a laser diode) generates a pulse of light, which activates a timer. When the light pulse strikes a target object, it is reflected back to a sensor, which is normally located near the laser diode, and the timer is turned off.

As the time (*t*) between the transmitted pulse and the reflected echo is known, the distance (*D*) to the target object may be calculated using a constant value of the speed of light (*c*):(1)$$D=\frac{ToF\times c}{2}$$

### 2.1. Principles and Components

LiDAR devices use laser light to determine distances and create detailed 3D maps of their surroundings [24,25,26]. A LiDAR sensor, which determines the precise distance from a target object by emitting a laser, includes a laser diode (LD), an avalanche photodiode (APD), a time-to-digital converter (TDC), and signal processing modules, as depicted in Figure 1. Therefore, these are the primary components of a LiDAR system, which are required for data capture, processing, and interpretation to measure distances and detect objects:

1. Laser: The color and intensity of a laser vary depending on the type of data gathered; nevertheless, every LiDAR payload includes a high-power laser [24,25,26]. Lasers in LiDAR systems produce pulses or continuous beams of light at certain wavelengths, usually in the near-infrared (NIR) range (905 or 1550 nm). Depending on the transmission method and application, these lasers can be either pulsed or continuous-wave (CW). Pulsed lasers are the most commonly used in LiDAR systems; they emit short, high-intensity bursts that last several nanoseconds. The distance to the target object is calculated by measuring the time-of-flight of each laser pulse. Due to their high instantaneous power, pulsed lasers are particularly accurate for measuring long distances and are frequently utilized in mapping, terrain modeling, and self-driving vehicles. CW LiDARs, by contrast, emit a continuous light beam and typically operate in one of two modes: amplitude-modulated continuous-wave (AMCW) or frequency-modulated continuous-wave (FMCW) [26]. AMCW systems measure distance using the phase difference between the transmitted and reflected signals. As they measure the phase shift of the transmitted and received lasers, they are generally less accurate and less suitable for long-range measurements [24,25,26]. FMCW LiDARs overcome these limitations by modulating the frequency of the continuous laser beam and calculating distance and velocity based on phase shifts induced by the Doppler effect. Although highly accurate, FMCW systems are more complex and less effective for long-distance detection due to frequency distortion caused by the Doppler shift.

2. Scanner/beam steering mechanism: The scanner directs and steers the laser beam through the surroundings while collecting the returned signals, defines the LiDAR coverage area, and impacts its resolution. Rotating mirrors and oscillating systems are used to control the laser beam. Section 2.2 explains the scanning mechanisms used in lidar sensors.

3. Receiver (photodetector): Avalanche photodiodes (APDs) and single-photon avalanche diodes (SPADs) are commonly used detectors that capture low-intensity reflected light signals—often with photon-level sensitivity—and convert them into electrical signals for further processing [24,25,26].

4. Signal processing unit: This LiDAR system component is responsible for turning the raw data collected by the photodetector into usable information such as distance measurements and 3D object mapping. It removes noise and irrelevant data captured by the photodetector and enhances detection accuracy by focusing only on the laser pulses reflected from objects of interest. It also boosts low-intensity return signals from small-scale or distant objects, such as drones, pedestrians, or small animals. These signals often have a low signal-to-noise ratio (SNR) due to limited reflectivity or long travel distance, which makes them difficult to distinguish from background noise. Amplifying such signals is critical in long-range LiDAR systems or low-reflectivity environments, where point cloud density may otherwise be insufficient for accurate detection and mapping [43,44,45]. After filtering and amplification, the unit estimates the time-of-flight (ToF) of each laser pulse. Finally, it transforms the processed data into a 3D point cloud representing the scanned environment, where each point indicates the location of a detected object. In some cases, the signal processing unit performs data compression and transmission tasks to manage large-scale LiDAR data—often exceeding 100 GB—in real-time applications [27,28,29].

5. Power supply unit: It is essential to supply the electrical energy necessary to power all components of the system and to ensure crucial operations such as powering the laser emitter, regulating the voltage for the receiver, enabling signal processing, controlling beam steering, and managing energy efficiency [27].

6. Positioning and orientation unit: This unit provides an accurate global position and orientation of the LiDAR sensor during data collection. It includes GPS/GNSS (Global Positioning System/Global Navigation Satellite System), IMU (Inertial Measurement Unit), INS (Inertial Navigation System), and SLAM (Simultaneous Localization and Mapping). The GPS/GNSS unit enables precise positioning, georeferencing, and trajectory tracking of the LiDAR system within a global coordinate system. The IMU unit ensures detailed information on the sensor’s orientation, acceleration, and movement, ensuring that the LiDAR data are correct even when the sensor is in motion or vibrating. The INS unit fuses data from the GPS/GNSS and IMU to produce continuous and highly precise position and orientation information, particularly in challenging environments or during dynamic motion. The SLAM unit enables the LiDAR sensor to simultaneously map and localize its surroundings in GPS-denied environments.

7. Control system unit: The control system coordinates all components of the LiDAR sensor. It handles laser emissions, receiver synchronization, data gathering, and filtering operations. It also interacts with other systems, such as GPS or IMU, and sets scanning settings and power management [28,29].

### 2.2. Classification and Mechanisms

LiDAR sensors may be classified depending on a number of factors, including scanning mechanisms, operational environments, measurement techniques, application areas, etc. LiDAR systems are categorized into non-scanning and scanning types based on their scanning mechanisms (see Figure 2). Scanning LiDARs use either mechanical or non-mechanical systems to sequentially steer the laser beam and scan the surrounding environment. In contrast, non-scanning LiDAR systems do not rely on any scanning mechanisms to move the laser beam across its field of view (FoV). Instead, they illuminate the full scene in a single light pulse and capture its reflection on a 2D sensor array, similar to a camera [30].

Mechanical scanning LiDAR sensors use physically moving components (rotating mirrors, oscillating prisms, entire sensor heads, etc.) to steer the laser beam across the environment and measure the distance to create accurate 3D maps. There are several types of mechanical scanning LiDAR: micro-electromechanical systems (MEMS) [30,31,32,33,34,46], optomechanical [35,36,37], and electromechanical [38] scanning LiDAR.

Mechanical MEMS scanning devices employ micro-mirror plates to generate precise scanning patterns and are classified as quasi-solid-state LiDAR because of their design, where the moving parts solely steer the laser beam in free space without involving the movement of optical components. MEMS-based scanning mirrors are generally classified as resonant or non-resonant [34]. Resonant MEMS mirrors operate at a constant resonant frequency to create regular oscillations for scanning, whereas non-resonant MEMS mirrors are more flexibly controlled by employing external actuators or controllers. These scanning mechanisms are widely used in various fields due to their compact size, affordability, low power consumption, and ability to perform high-resolution scans [46]. They play a crucial role in applications such as autonomous vehicles, where they enhance obstacle detection and navigation; robotics, where they support accurate spatial mapping; space exploration, benefiting from their lightweight and energy-efficient design; medical imaging, where they facilitate the development of compact, high-resolution imaging tools; and mobile applications, where their portability and efficiency are vital. Despite these advantages, MEMS mirrors face specific limitations, including a restricted range and field of view (FoV), which may constrain their applicability in wide-area scanning tasks. In addition, they are sensitive to environmental factors such as temperature changes and mechanical vibrations, which can affect their resonant frequency and overall performance [30,31,32,33,34]. These challenges highlight the need for careful consideration when deploying MEMS systems in demanding environments.

Optomechanical scanning systems often use rotating polygon mirrors and Risley prisms that physically rotate to steer the laser beam over the scene being scanned. These systems are perfect for automobile LiDAR, remote sensing, and even biological imaging since they can maintain accurate beam control over long durations, as well as achieve a high field of view (FoV) and range. Optomechanical systems are extremely effective, but they are heavy and feature moving components that can wear out over time, making them less suitable for small or lightweight applications than MEMS-based equivalents. However, in cases that require high power and long-range capabilities, optomechanical mechanisms continue to be the dominant choice [35,36,37].

Electromechanical scanning devices employ revolving or oscillating mirrors driven by electric motors (such as stepper motors or servo motors) to deflect a laser beam in certain directions. This enables the LiDAR system to perform 2D or 3D scanning on broad regions. These systems are among the first and most extensively used scanning mechanisms in LiDAR because of their simplicity, efficacy in attaining high angular resolution, a large horizontal and vertical field of view, high accuracy, and long-range capabilities. These systems are widely used in self-driving cars, infrastructure monitoring, environmental mapping, remote sensing, and atmospheric studies. Although electromechanical systems are strong and reliable, they do have certain limitations. Moving parts are prone to wear and tear, which could jeopardize long-term reliability. Furthermore, these systems tend to be bulkier and heavier compared with newer, more compact options such as MEMS or solid-state LiDAR, which offer comparable capabilities without moving components [35,38].

The non-mechanical scanning LiDAR sensor is also known as solid-state beam scanning since it has no moving parts such as rotating mirrors or prisms. Solid-state beam scanning LiDAR systems frequently employ optical phased arrays (OPAs) to steer the laser beams by altering the phase of light emitted at various spots in the array [39,40]. OPA scanning techniques in LiDAR systems are gaining popularity because of their non-mechanical beam steering capabilities and are regarded as being fully solid-state, with no moving components. They offer the potential for very quick scanning while being highly precise in tuning for varied beam directions. Recent research [39] focuses on advances in photonic integrated circuits (PICs) for OPAs, which improve their compactness, speed, and energy efficiency. These integrated circuits have fast response times and compatibility with existing manufacturing processes (such as CMOS), making them suitable for mass production in sectors like automotive LiDAR. Another OPA-based LiDAR research [39] presented a target-adaptive scanning method that aims to adjust the scanning resolution depending on a detected target. Based on this method, high-resolution scanning is retained for only essential objects, while unimportant background regions are scanned coarsely. Therefore, this strategy increases efficiency as well as the system’s ability to focus on critical aspects in real-time applications such as autonomous driving [40]. Although OPA technology has great advantages, such as no moving components, quick beam steering, compact size, and energy efficiency, it faces challenges in managing beam width and reducing grating lobes [39,40].

Non-scanning LiDAR is commonly known as flash LiDAR. The flash LiDAR system employs a broad laser beam and a huge photodetector array to gather 3D data in a single shot, making it perfect for real-time applications. Recent studies [41,42] have focused on the integration of SPAD (single photon avalanche diodes) sensors with Flash LiDAR systems, which allows for increased sensitivity and performance in low-light conditions. SPAD-based flash LiDARs can operate over long distances (up to 50 km) and are being studied for precision landing systems for planetary missions. Therefore, flash LiDAR offers primary advantages such as the lack of moving components, quick data capture, and the ability to create high-resolution 3D maps in real time, making it an invaluable tool for terrestrial and space applications. However, because of their limits in range and resolution compared with scanning LiDAR systems, they are usually used in short-range applications [41,42]. The differences between these LiDAR scanning mechanisms can be distinguished according to the brief comparative information in Table 2 below.

Based on the dimension of acquired data, LiDAR systems that can collect spatial information come in three varieties:-One-dimensional (1D) LiDAR [47,48];-Two-dimensional (2D) LiDAR [49,50,51,52,53];-Three-dimensional (3D) LiDAR [54,55].

As mentioned above in Section 2.1, the LiDAR system has several basic components, such as a laser, scanner, receiver, signal processing unit, etc. However, the LiDAR system can still operate without a scanning mechanism and is known as 1D LiDAR. 1D LiDAR mostly estimates the distance along a single fixed axis or direction. In addition, its laser is fixed and directed in one direction rather than rotating or sweeping over a larger region. In [47], a low-cost, high-precision pointing mechanism for obstacle detection on railway tracks was demonstrated over long distances using a gimbaling platform coupled with a 1D LiDAR sensor. The gimbal enables the LiDAR sensor to scan a large area dynamically, detecting obstructions such as animals, debris, and equipment on rails to prevent accidents. The actual pointing accuracy of the system was assessed by conducting a controlled, indoor, and long-distance experiment. The experiment findings showed that the system can reliably target individual points over large distances, with angular resolution enough to detect humans at 1500 m using long-range LiDAR. In general, the study contributes significantly to transportation safety and provides a solid platform for future advances in obstacle detection technologies. Rather than using expensive and sophisticated 3D LiDAR systems, the authors of [48] developed an inexpensive indoor navigation system capable of mapping static environments and detecting objects. The proposed system uses a 1D LiDAR sensor to scan its surroundings and create 3D maps by combining data from many scans captured over time as the LiDAR-equipped vehicle moves across the surroundings. The prototype of the system is constructed with a LiDAR Litev3 sensor, two servomotors, and a pan-tilt mechanism and is designed primarily for tiny autonomous bots or automated guided vehicles (AGVs) when vehicle speed is not a concern. It assumes that the environment is static relative to the sensor, with only obstructions being relevant. The prototype determines obstruction coordinates by scanning 150° horizontally and 120° vertically using a fast scanning approach. Once the coordinates are spotted, the sensor focuses on the obstacle to generate a detailed map, which allows the vehicle to analyze the object’s profile and provide navigation instructions for obstacle avoidance. Based on the results of the experiment, obstacles within a 1-m range are successfully detected, followed by the creation of an object profile. The use of an adaptive scanning method reduced scan time by more than half and recognized the object’s presence and shape as rapidly as possible.

Recent research works have explored various aspects of 2D LiDAR technology in UAV detection and tracking. For example, in [49], the researchers have examined 2D LiDAR-based UAV detection and presented a formulation for the probability of detection in various settings using a LiDAR-turret system. The proposed system relies on sparse detections instead of dense point clouds, as well as estimates of motion and active tracking. The authors offered the theoretical framework for analyzing the performance of a 2D LiDAR-based detection system and better understanding its limitations. The work consists of field experiments involving the detection of multiple different-sized drones based on a modern LiDAR system and highlights its effectiveness in UAV identification and tracking using sparse data. A LiDAR-assisted UAV exploration algorithm (LAEA) for unknown surroundings was proposed in [50]. The proposed system framework has three major modules: map construction, target selection, and motion planning. The approach uses 2D ToF (time of flight) LiDAR and a depth camera to swiftly capture contour information from the undiscovered surroundings, then combines data from two sensors to generate a hybrid 2D map. By using fused multi-sensor data, the map construction module produces both high- and low-resolution 3D occupancy maps, as well as 2D occupancy maps for detecting special frontier clusters. The target selection module is responsible for frontier-based viewpoint generation, detection of tiny and isolated frontier clusters, and solution of the asymmetric traveling salesman problem (ATSP). Finally, the motion planning module then conducts specific azimuthal trajectory optimization utilizing the EIG (environmental information gain) optimization approach, resulting in a safe trajectory that allows the UAV to collect more information. The authors evaluated the suggested algorithm’s efficacy using the Gazebo simulator, comparing it to cutting-edge solutions such as FUEL (Fast UAV Exploration) and FAEP (Fast Autonomous Exploration Planner). The simulation results revealed that the proposed LAEA approach outperforms those two methods in terms of flight distance and exploration time. The feasibility of the proposed approach was validated on a robotic platform outfitted with an RGB-D camera, 2D LiDAR, and an Nvidia onboard computer in two distinct real-world scenarios (indoors and outdoors). A robust short obstruction and pothole detection and classification approach using a cost-effective 2D LiDAR sensor mounted on a mobile platform was proposed in [51]. As the research goal was to detect short obstacles in the ground, the LiDAR sensor was placed downward-looking. The data were acquired using a Hokuyo UBG-04LX-F01 2D LiDAR sensor (Hokuyo Automatic Co., Ltd., Osaka, Japan). Then, the acquired data are converted into a point cloud, which is a straightforward transition from polar coordinates to Cartesian. To identify obstacles, the point cloud is segmented into lines, and based on their average height, the lines were classified as either ground, pothole, or obstacle. The experimental findings showed that the suggested method properly recognizes obstructions and potholes in a structural environment. Nevertheless, the point cloud is over-segmented with unnecessary lines, which might be addressed by adjusting the line refinement parameter. The authors plan to consider the dynamic objects and analyze their movement in the lines in their future study. Reference [52] presents an effective moving object recognition approach based on 2D LiDAR and frame-to-frame scan matching. The researchers used an autonomous mobile robotic system (MRS) with an LD-OEM 1000 LiDAR to test the suggested SegMatch algorithm. This configuration was used to ensure collision-free passage across a mapped region. The autonomous MRS is controlled with an Application Programming Interface (API) and communicates via LoRa technology. As well as a WiFi router was installed that connects with the LiDAR and an external processing unit via TCP/IP (Transmission Control Protocol/Internet Protocol). The main objective of the autonomous MRS is to execute active SLAM (simultaneous localization and mapping). The authors evaluated the proposed approach for dynamic object detection in three scenarios: stationary measurements, SLAM execution, and real-time measurements. For these purposes, the necessary data were obtained via a LiDAR scanner and then fed into the SLAM algorithm, which calculated deviations and created a 2D map. Then data preprocessing was carried out to make data appropriate for point cloud generation. Afterward, the proposed method was applied to processed data to detect dynamic objects. When detecting dynamic objects, the authors encountered problems such as the occurrence of defects on the resulting map and probable collisions of the autonomous MRS with the dynamic object. The algorithm’s advantages include a rapid response time and the ability to be employed in heavily populated areas. To gain a full perspective of the environment, the proposed approach might be enhanced with an ITS (Intelligent Transport System) architecture-based multi-agent system using several mobile robotic systems. Another study in [53] presents a comprehensive analytical formalism focused on 2D LiDAR structured data representation, object detection, and localization within the realm of mobile robotics. The authors described a formalized approach for LiDAR data processing and its mathematical representation, converting raw sensor data into intelligible representations suited for a variety of robotic applications. The proposed analytical formalism includes advanced algorithms for noise reduction, feature extraction, and pattern recognition, allowing the mobile robot to detect static and dynamic objects in its proximity. The efficacy of the proposed method was validated by conducting numerous experiments in a scenario with items of various configurations, sizes, and forms that properly simulate a real-world use case. The experiment outcomes demonstrated that the suggested technique can efficiently recognize and separate objects in semi-structured environments in under 50 milliseconds. Furthermore, the authors are sure that the mathematical framework’s simplicity provides minimal computing effort and efficiency, establishing the groundwork for innovative solutions in a wide range of cases. In their future work, the authors plan to study the implementation of machine learning techniques into the suggested framework for object recognition and classification tasks.

In [54], the authors studied the performance of a 3D LiDAR system for UAV detection and tracking, concentrating on the robustness of effective range estimation to various drone types and shapes, as well as the visibility robustness to environmental and illumination conditions. Additionally, the potential of 3D LiDAR for UAV detection and tracking under lighting conditions was evaluated. The effective range estimation experiment was carried out by mounting a Livox Mid-40 3D LiDAR (Livox Technology Co., Ltd., Hong Kong, China) in an open field and flying two black and white UAVs at distances up to 80 m. Experiment results indicated that the color of the UAV had a considerable effect on its reflectivity and, hence, the range of detection, with a smaller white UAV being visible at a greater distance than a bigger black UAV, despite the latter’s size advantage. Since the white UAV had a greater detection range, the visibility robustness experiment was carried out only with one UAV during three distinct time periods of the day with the same background. The experiment outcomes revealed that the number of captured LiDAR points and reflectivity decreases as distance increases; however, at extreme distances, mean reflectivity increases due to the dominance of the UAV’s most reflective parts. This means that UAV localization remains reliable even in low-light conditions without a reduction in detection range. Also, a 3D UAV tracking experiment was performed by continuous LiDAR scanning at three different time periods of the day to track the white UAV’s motion and assess its trajectory within the scan duration. The experiment outcomes demonstrated that UAV trajectory tracking remains effective across different lighting conditions and UAV speeds, showcasing the potential of 3D LiDAR for robust UAV tracking tasks. Preliminary research indicates that 3D LiDAR has considerable potential for robust UAV detection, localization, and tracking. Future research directions include extending the detection range to 200 m without increasing laser power, developing a mobile system to track high-speed UAVs without sacrificing point cloud density, and developing a real-time system that integrates AI and machine learning for UAV detection and tracking based on different shapes, materials, and reflectivity. The research also intends to follow drone swarms in 3D and evaluate LiDAR’s performance in harsh weather conditions such as snow, fog, and rain. The authors of [55] proposed a novel approach for multi-object tracking (MOT) in 3D LiDAR point clouds by combining short-term and long-term relations to enhance object tracking over time. The short-term relation analyzes geometrical information between detections and predictions and exploits the fact that objects move gradually between consecutive frames. The long-term relation, on the other hand, considers the historical trajectory of tracks to assess the degree to which a long-term trajectory matches the present detection. An effective Graph Convolutional Network (GCN) approach was used to assess the matching between the present detection and existing object trajectories. By representing the relations between identified objects as a graph, the system enhances its capacity to associate objects across frames, especially in dense or cluttered settings. In addition, an inactive track was kept to solve the issue of incorrect ID switching for objects that had been occluded for an extended duration. This method allows the system to keep object tracks more reliably, especially under challenging scenes, such as when several objects move close together or when partial occlusion occurs. Therefore, the proposed solution with a multi-level association mechanism successfully mitigates problems such as ID switching following occlusion, leading to increased tracking accuracy and resilience in complex environments. However, the work might benefit from further investigation into its computing efficiency and applicability to a wider range of applications. Despite these limitations, the comparison with cutting-edge LiDAR and LiDAR-camera fusion tracking systems revealed the proposed approach’s clear efficacy in improving robustness, notably in resolving ID switching and fragment issues.

Each type of LiDAR sensor has unique benefits and drawbacks in terms of range, size, power consumption, resolution, and price. The sensor selection depends on the application’s unique needs, such as detection range, accuracy, ambient conditions, and budget.

## 3. LiDAR Data Processing Techniques

### 3.1. Clustering-Based Approaches for LiDAR Data Processing

Recently, it has been brought into the field of 3D point-cloud processing, considerably boosting the ability to analyze and interpret point-cloud data. Because of its simplicity and efficacy, clustering-based object detection has been widely used in the field of LiDAR data processing. Euclidean Distance Clustering (EDC) [56] is one of the most fundamental and widely used clustering algorithms for LiDAR data. This approach groups points that are close to each other within a given distance threshold and groups them based on proximity. The Density-Based Spatial Clustering of Applications with Noise (DBSCAN) [57] method is another common clustering technique for LiDAR-based object detection. DBSCAN groups points according to their local density, allowing it to deal with noisy data and identify clusters of various shapes. A novel real-time obstacle detection and navigation approach for UAVs using 2D LiDAR technology was presented in [58]. The proposed obstacle detection system is considered to be lightweight and cost-effective by consisting of a 2D LiDAR sensor RPLIDAR A2 (Slamtec, Shanghai, China) and a Raspberry Pi 3 Model B (Raspberry Pi Foundation, Cambridge, UK). Additionally, the DJI Matrice 100 serves (DJI, Shenzhen, China) as a flight platform. The employed 2D LiDAR consists of two main components: a fixed mount attached to the UAV and a rotating element that enables 360-degree environmental scanning. The detection method includes point cloud correction and the clustering algorithm based on relative distance and density (CBRDD). Because the LiDAR sensor is mounted on a moving drone, it continuously scans the environment relative to the drone’s changing position and orientation. Therefore, the point cloud obtained from the LiDAR is affected by the motion, and there is a difference between the ideal and actual point clouds, which necessitates the correction of obtained point clouds before putting them into practice. Point cloud correction was performed by converting the point cloud series from a polar to a Cartesian coordinate system and then estimating the UAV position based on the IMU (Inertial Measurement Unit), which refers to a velocity estimation model. The relative distance between two adjacent points and the density distribution are the main features to be extracted. Then, the CBRDD clustering algorithm was applied to these features to obtain the distribution information of obstructions. The experimental part includes both simulation and actual experiments to validate the proposed approach. The DBSCAN (Density-Based Spatial Clustering of Applications with Noise) algorithm was chosen for the comparison. Based on the simulation experiment outcomes, the CBRDD clustering algorithm outperformed DBSCAN for the uneven-density point clouds. While in the actual experiment for uniform point clouds, they showed the same result. In [59], the authors offered a robust UAV tracking and position estimation system employing fused data from Livox Avia and LiDAR 360 sensors and a clustering-based learning detection (CL-Det) approach. LiDAR 360 used 3D point cloud data to give 360-degree coverage. Livox Avia, on the other hand, provided focused 3D point cloud data for a specific timestamp, which often represents the origin point or drone location. Initially, the timestamps of these two sensors are aligned to ensure temporal coherence. Then, fused LiDAR data coordinates were compared with the drone’s known ground truth positions at the respective timestamps. Furthermore, the CL-Det clustering method, particularly DBSCAN, was applied to process LiDAR data and isolate UAV-related data from the environment points. The DBSCAN method clustered the UAV-related point clouds and selected the object of interest, assuming that the biggest non-environment cluster represents the UAV. The UAV’s position is estimated using the mean of the selected point cloud cluster, expressed in Cartesian coordinates (x, y, z). To address the issue of sparse LiDAR data, the authors used historical estimations to fill in the data gaps, maintaining continuity and accuracy in UAV tracking even when certain LiDAR measurements were unavailable. Therefore, the proposed multi-sensor integration and clustering-based UAV detection and tracking system demonstrates the potential for real-time, precise UAV tracking, even in sparse data conditions.

Clustering-based LiDAR object detection techniques are effective; however, they have some drawbacks. Usually, these methods rely on manually tuned parameters, such as distance thresholds, which must be adjusted for various environments, limiting their applicability. These approaches may also suffer in complex or cluttered environments, resulting in excessive or insufficient segmentation. Furthermore, clustering-based techniques frequently rely on handmade features, making them less successful in complex scenarios than deep learning models that automatically learn features. However, hybrid approaches that integrate clustering with machine learning or deep learning detection methods show promise in overcoming these issues. Deep learning technology has demonstrated its superiority in image, audio, and text processing.

### 3.2. DL-Based Approaches for LiDAR Data Processing

Researchers are now turning to deep learning algorithms to achieve comparable advances in 3D object recognition and classification with LiDAR data. By enhancing the accuracy, efficiency, and durability of 3D object identification systems, these innovations are defining the future of a variety of sectors, including autonomous vehicles, robotics, aerial surveying, mapping, etc. The authors of [60] introduced an innovative airborne LiDAR-based solution to detect and localize drone swarms using deep learning in 3D by modifying and embedding the PointPillars neural network to detect airborne objects. The PointPillars deep learning model was modified by horizontally dividing the LiDAR FoV into distinct layers, with anchor boxes assigned to each layer, placing the center of each anchor box in the center of its corresponding layer. A scenario-based digital twin was used to replicate close encounters and critical safety events, which were then used to generate training data. Data augmentation was carried out by supplementing real-world data with high-quality synthetic drone data, which served to increase the accuracy and efficiency of both training and inference processes. The efficiency of the proposed method has been evaluated in real-world datasets using primary evaluation metrics, and it has demonstrated notable performance, achieving 80% recall and 96% precision. In [61], the authors presented a novel approach for indoor human localization and activity recognition (HAR) based on an autonomous moving robot equipped with 2D LiDAR that could be useful in drone detection. The first step of the proposed method relied on collecting data points based on continuous movement, transforming them into absolute coordinates, cleaning the scan, removing noisy data points, identifying the person’s associated data points, and determining their location. In the next step, human activities are recognized based on the detected subject’s related data points. The preprocessed and interpolated data points were then trained using three variations of a convolutional long-short-term memory (LSTM) neural network to classify nine types of human activities, such as running, standing, walking, falling down, etc. The proposed LSTM models’ architecture is quite simplistic, with only a few layers. The simulation results were compared with traditional approaches that employ static 2D LiDARs mounted on the ground and proved the efficacy of the proposed approach. The proposed approach outperformed conventional methods in detecting falls and lying body actions with an accuracy of 81.2% and 99%, respectively.

### 3.3. Real-World Applications of LiDAR with Deep Learning

In addition, the merging of deep learning algorithms with LiDAR technology has led to substantial breakthroughs in drone detection by identifying and monitoring drones in airspace for security and safety purposes. Recent breakthroughs in deep learning have considerably improved the capabilities of LiDAR-based systems for a variety of UAV-related and environmental perception applications. A review of selected real-world case studies demonstrates the practical use of DL algorithms in a variety of operational scenarios. For example, the PointPillars DNN architecture has been successfully used for swarm self-noise filtering in LiDAR point clouds with high percentages and numbers of removed points [60]. Similarly, the fusion of the DL-based YOLOv2 algorithm with Kalman filtering has been utilized for UAV detection and 3D tracking in BVLOS (Beyond Visual Line of Sight) [62]. More sophisticated fusion solutions, such as the use of YOLOv9 with fisheye cameras, LiDAR, and mmWave radar, have improved 3D tracking in outdoor UAV applications [63]. In indoor settings without GNSS signals, approaches such as Conv-LSTM have achieved over 99% accuracy in motion recognition using 2D LiDAR [61], while Ouster LiDAR-based signal image fusion has proved useful for accurate UAV tracking [64]. Beyond UAV detection, LiDAR and the GoogLeNet CNN-based frameworks have been effectively used in other fields, such as geological engineering. Ge et al. [65] accurately identified rock joints in 3D point clouds and determined their orientation. The proposed model obtained high classification accuracy while simply utilizing point coordinates and normals as input, and it was validated on two real-world datasets. The trained model identified three natural discontinuity groups, and individual discontinuities were extracted using the density-based spatial clustering of applications with noise (DBSCAN) algorithm. Their orientations were then determined using the principal component analysis (PCA) algorithm. In the realm of remote sensing, lightweight models such as hierarchical graph networks (HGNets) and ConvPoint have produced impressive results in 3D object detection on multi-platform LiDAR datasets [66]. Meanwhile, in autonomous driving applications, GNN based on the neighbor feature alignment mechanism showed high 3D object detection and location performance in terms of three detection difficulty cases (easy, moderate, and hard) on the KITTI benchmark [67]. Collectively, these research works (see Table 3) demonstrate that DL approaches not only improve the accuracy and reliability of LiDAR-based detection systems but also broaden their use in autonomous driving, geological engineering, drone swarm noise filtering, and human activity recognition (HAR). The constant integration of LiDAR with complementary sensor modalities, as well as the use of benchmark datasets, emphasizes the approaches’ practical relevance and scalability.

The following Table 4 provides a detailed comparison of important deep learning algorithms for LiDAR data processing by emphasizing data representations, major techniques, advantages, and limitations.

### 3.4. Sensor Fusion for Enhanced UAV Detection

The challenges discussed above with drone detection based on LiDAR data demonstrate the benefit of employing several sensors rather than a single sensor in object detection. Combining LiDAR and camera data enhances UAV detection. LiDAR provides exact 3D spatial data, while cameras collect rich visual detail. By combining both technologies, the system addresses the limitations of each sensor: LiDAR’s weather sensitivity and the camera’s low performance in low-light circumstances. The end result is a more robust and dependable UAV detection system. Therefore, the study [76] attempted to fuse the strengths of these two sensors to reach accurate detection of static and dynamic obstacles, making it particularly useful for UAV navigation in complex environments. Because the calibration of two sensors is required prior to integrating their data, the suggested system architecture first calibrates the LiDAR and camera sensors offline. The joint calibration results in a series of incomplete and sparsely distributed unordered discrete points that require a suitable point cloud segmentation technique for their processing. Point cloud segmentation is used to assign discrete points to detected obstacles and group points from the same obstacle into one group to assess their spatial distribution. Furthermore, fusion detection algorithms are designed for linear and surface static and dynamic obstacles. To verify the effectiveness of the proposed fusion-based obstacle detection algorithms, the authors gathered obstacle data from a UAV platform equipped with LiDAR and camera sensors, focusing on scenarios such as power line avoidance, building avoidance, and encounters with tiny low-altitude UAVs in low-altitude suburban areas. The detection results for different obstacles were evaluated using the over-segmentation rate and Intersection over Union (IOU) values. The comparative data verification analysis revealed that the proposed LiDAR-camera fusion-based detection algorithm beats solo LiDAR in various real-world motion scenarios, significantly increasing the distance and size detection accuracy of static obstacles, as well as the state estimation accuracy of dynamic obstacles. Another research [77] proposes a potential multi-sensory strategy for detecting and classifying small flying objects such as drones by combining data from several sensors. This approach uses a mobile multi-sensor platform equipped with two 360° LiDAR scanners and pan-and-tilt cameras in both the visible and thermal IR spectrum. Based on the proposed approach, the multi-sensory system first detects and tracks flying objects using 3D LiDAR data, after which both IR and visible cameras are automatically directed to the object’s position to capture 2D images. A CNN is then applied to detect the region of interest (ROI) and classify the flying object as one of eight types of UAVs or birds. The suggested multi-sensor fusion ensures more robust detection by integrating the characteristics of the above technologies, as radar sensors provide long-range detection, LiDAR gives high-resolution 3D spatial information, and cameras provide optical and infrared data for accurate object detection. The first multimodal UAV classification and 3D pose estimation algorithm for an accurate and robust anti-UAV system was presented in [63]. The proposed multimodal anti-UAV network includes UAV-type classification and 3D tracking pipelines. The UAV-type classification pipeline is mostly based on image data, whereas the UAV 3D tracking (pose estimation) pipeline is primarily based on LiDAR and radar data. The dataset of four types of UAVs was gathered from a stereo fisheye camera, conic and peripheral 3D LiDAR, and mmWave radar sensors. For better performance, the classification pipeline uses sequence fusion based on feature similarity, ROI cropping, and keyframe selection using YOLOv9. The pose estimation pipeline includes dynamic point analysis, a multi-object tracking system, and trajectory completion. The classification pipeline effectively combines information across sequences by applying a soft vote strategy, which improves the UAV type detection accuracy. Point cloud-based UAV detection was evaluated using primary evaluation metrics. Noise detections and missed trajectories are corrected using the multiple-object tracker. Recent advances in multi-sensor fusion have shown that the integration of several sensor modalities, such as LiDAR, infrared (IR), visible cameras, and acoustic sensors, may considerably improve the robustness and accuracy of UAV detection systems [78]. Each sensor provides complementary data: LiDAR provides high-resolution spatial and depth information, infrared sensors capture thermal signatures useful in low-light or obscured conditions, video cameras provide rich visual context, and acoustic sensors detect UAV-specific sound patterns, particularly in scenarios where visual line-of-sight is limited. Overall, combining multimodal data with modern DL algorithms produces reliable UAV detection and classification results. However, the issues associated with sensor alignment and calibration, as well as the large computing resources, necessitate additional exploration.

#### 3.4.1. Integrating LiDAR for Robust UAV Detection in GNSS-Denied Environments

LiDAR technology has proven to be a highly successful technique for detecting UAVs in GNSS-denied zones, where regular GPS systems may be inaccurate or not available. LiDAR provides precise spatial data and real-time mapping, allowing for robust detection, tracking, and navigation capabilities even in challenging environments such as urban regions, forests, or indoor places. Several studies [62,64,79,80] have investigated the use of LiDAR for UAV detection and tracking in GNSS-denied environments, with an emphasis on increasing resilience using sophisticated algorithms and sensor fusion approaches. Park et al. [62] proposed a novel sensor fusion-based airborne object detection and position estimation technique for UAV flight safety in BVLOS (Beyond Visual Line of Sight). To detect aerial objects in images captured by a vision sensor, CNN-based YOLOv2 architecture was employed, while a clustering algorithm was used to detect objects from LiDAR point cloud data. Further, to improve the detection accuracy, a Kalman filter was used to integrate these two different sensor data based on multiple estimated state fusion approach. The Kalman filter leveraged two- and three-dimensional constant acceleration models for estimating the states of vision and LiDAR sensors, respectively. The 3D position of the detected object was determined using the depth of LiDAR and the image’s center point, which is the outcome of the fusion method. The suggested approach was validated using simulations in the Gazebo simulator to provide a realistic flight scenario. Based on simulation results, in comparison to individual camera or LiDAR systems, combining 3D spatial data from LiDAR with visual information significantly increased detection speed and accuracy, as well as decreased false positives. In [64], the authors investigated the potential of LiDAR as a camera sensor for real-time UAV tracking. They proposed a novel approach to UAV navigation in GNSS-disabled areas based on the combination of signal images and dense 3D point cloud data captured by an Ouster LiDAR sensor. The three distinct data sequences were captured in indoors with the distance between the LiDAR and the UAV ranging from 0.5 m to 8 m. The UAV tracking process involved two basic steps: initializing the UAV’s position and fusing image signals with dense point cloud data. The absolute pose error (APE) and velocity error were calculated using ground truth data from the MOCAP (motion capture) system to validate the UAV’s estimated pose and velocity accuracy. The proposed method was compared against a UAV tracking approach that relies exclusively on either Ouster LiDAR images or point clouds, and the experiment results of the proposed one outperformed the solo models. To extend application areas and enhance detection accuracy, the authors’ future work entails the integration of Ouster LiDAR images, point clouds, and standard RGB images. Another novel multi-sensory solution with several LiDAR sensors for accurate UAV tracking in GNSS-denied environments was presented in [79]. The authors also proposed a novel multi-LiDAR dataset particularly designed for multi-UAV tracking captured from a 3D spinning and two low-cost solid-state LiDAR sensors with various scan patterns and FoV, as well as an RGB-D camera. The proposed dataset includes UAV data with different sizes ranging from micro-aerial vehicles to standard commercial UAV platforms taken from both indoor and outdoor environments. Sensor calibration was performed by determining extrinsic parameters, aligning the point clouds of each sensor to the reference frame, optimizing the relative transformation between the reference frame and the LiDARs, etc. The MOCAP system was then used to generate accurate ground-truth data. Based on environmental scenarios and trajectory patterns, the dataset was organized into structured and unstructured indoor categories as well as unstructured outdoor categories. Tracking performance was evaluated using the root mean square error (RMSE) metric. In general, the research emphasizes the importance of addressing UAV tracking in GNSS-denied areas and reveals that multi-LiDAR systems may greatly improve tracking accuracy and reliability. However, it suggests future research into the computational issues involved with such systems, as well as the possible integration of additional sensor modalities to increase robustness in diverse conditions. [80] developed a robust architecture for UAV navigation in a GNSS-denied environment that integrates LiDAR, camera, and IMU sensors into a single odometry system. The proposed Resilient LiDAR-Visual-Inertial Odometry (R-LVIO) system aims to decrease trajectory error and enable reliable UAV operation by estimating the UAV’s state and creating a surrounding map. In addition, the system framework employs robust pose estimation approaches, including hybrid point cloud registration and visual feature depth cross-validation. The extrinsic parameters of three distinct sensors are calibrated to a single coordinate system. The IMU frame served as the primary coordinate system, while the camera and LiDAR acted as sub coordinate systems. Gaussian probability-based uncertainty was employed to represent irregular surfaces in unstructured environments. This uncertainty is then separated into eigenvalues and eigenvectors, and a pose estimation goal function is developed to achieve precise localization. In addition, unstructured hierarchical environments are utilized to evaluate the localization accuracy of the proposed system. The proposed system is built to manage real-time processing and recover from individual sensor failure, making it ideal for dynamic and difficult GNSS-denied environments. In terms of precision and resilience, the experimental results outperformed previous approaches.

#### 3.4.2. Challenges in Sensor Fusion-Based UAV Detection

Sensor fusion approaches, especially those that combine LiDAR with complementing modalities such as RGB cameras, radar, and IMUs, have the potential to significantly increase the robustness and precision of UAV detection. However, actual application in real-world settings presents various ongoing challenges, as detailed below Table 5.

## 4. Discussion

Due to the continuous improvement in the technology of unmanned aerial vehicles, the scope of use of drones is also increasing rapidly. Their ability to carry various loads is the reason for the frequent occurrence of drone incidents. This, in turn, emphasizes the relevance of a robust and effective drone detection system. This review paper begins with an introduction section, which highlights the comparative description of the advantages and downsides of traditional anti-drone systems (see Table 1).

The fundamentals of LiDAR sensors are explained by describing the structure and principle of a typical LiDAR sensor, as well as its components, in Section 2. Furthermore, the comprehensive LiDAR classification study gives an overview of the most recent developments in LiDAR scanning mechanisms. Figure 1 shows the detailed classification of scanning and non-scanning LiDAR sensors. Table 2 provides brief comparative information on LiDAR sensors based on these scanning mechanisms. Based on the dimension of acquired data, 1D, 2D, and 3D LiDAR sensors were also discussed. Related works on recent LiDAR scanning mechanisms proved that optomechanical [35,36,37], MEMS [30,31,32,33,34], and flash [41,42] LiDAR are the most used LiDAR scanning methods, each with its unique advantages and limitations. Therefore, the appropriate LiDAR scanning mechanism must strike a compromise between range, resolution, cost, and application requirements.

Deep learning approaches to LiDAR data processing [81] have advanced dramatically, allowing for the effective handling of unstructured 3D point clouds for a variety of applications, including object detection, segmentation, and tracking (see Table 3). Section 3 presents a comparative analysis of key deep learning approaches used for LiDAR data processing by highlighting their data representations and main techniques, as well as advantages and limitations, in Table 4. These approaches differ in terms of how they deal with 3D data, computational complexity, and their applicability for highly accurate or real-time tasks [82]. For example, point-based methods [68,69], such as PointNet, are simple and successful in classifying point clouds because they directly process point clouds and capture features using shared MLPs. Although voxel-based approaches [70,71], such as VoxelNet, convert the sparse and irregular point clouds into a volumetric 3D grid of voxels and are more suitable for detecting large-scale objects. Projection-based methods [72,73] convert 3D LiDAR point clouds into 2D views, sacrificing some 3D detail for computational efficiency, whereas graph-based approaches [66,67] excel at capturing complex spatial relationships but require more computational power. Hybrid methods [74,75] integrate the benefits of many methods, although they are frequently more complex.

State-of-the-art studies on LiDAR-based UAV detection and tracking using clustering-based and deep learning-based approaches have been discussed in Section 3. There are several clustering or segmentation techniques, such as EDC (Euclidean Distance Clustering), k-means, and DBSCAN. Some research works on UAV point cloud classification and pose estimation based on density-based spatial clustering of applications with noise (DBSCAN) methods were considered [58,59]. Several works that used the deep learning-based UAV detection and tracking method were exposed in [60,61]. Furthermore, different challenges, such as weather conditions, including rain, fog, and snow, can affect the performance of LiDAR by adding noise and decreasing the detection accuracy (see Table 5). Research into multi-modal sensor fusion [83], which merges LiDAR with other sensor modalities, has shown promise in solving some of these issues. Therefore, the benefit of employing several sensors rather than one sensor for the detection and tracking task of UAVs was considered later in [62,63,76,77]. The ability of LiDAR technology to provide accurate spatial data and real-time mapping, allowing robust UAV detection and tracking capabilities even in challenging GNNS-denied zones, was presented in [64,79,80].

## 5. Conclusions and Future Works

Although the fusion of LiDAR and acoustic sensors is still a developing field, recent research such as Semenyuk et al. [78] sees this integration as a potential path for enhancing UAV detection robustness in noisy and GNSS-denied environments. They emphasize the possibility of merging audio characteristics with LiDAR-based spatial data, employing sophisticated sensor fusion and deep learning techniques [81].

In conclusion, LiDAR has shown to be an important technique for UAV identification, providing exact 3D spatial data even in complex and challenging contexts. This paper discussed the evolution of LiDAR systems, their integration with sophisticated deep learning models, and their use in real-world applications such as rock joint analysis, autonomous driving, and GNSS-denied UAV tracking.

Deep learning algorithms, particularly hybrid and graph-based models, offer more precise and adaptive detection from LiDAR point clouds [82]. Real-world findings show considerable improvements in accuracy, robustness, and flexibility across a range of application cases. Meanwhile, sensor fusion systems [83] that combine LiDAR with cameras, radar, or IMU improve detection robustness while posing calibration, synchronization, and data volume challenges.

Moving forward, advancement will be dependent on the creation of unified multi-sensor datasets, real-time processing algorithms, and robust detection frameworks. Optimization methods for machine learning models, such as those suggested by Smailov et al. [84], will be crucial in improving detection accuracy. The combination of LiDAR and AI continues to reshape airborne danger detection, opening the door for smarter and safer autonomous systems.

## Figures and Tables

**Figure 1 sensors-25-02757-f001:**
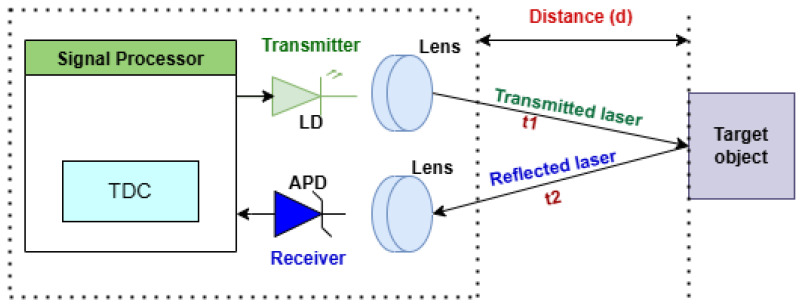
Structure and principle of a typical LiDAR sensor. The LD sends out the laser, which is focused by a light-transmitting lens; then, the emitted laser is reflected back from the target object and received by the APD via the light-receiving lens. Further, the TDC measures the subtraction between the time the LD sends the laser and the time the APD receives it and converts the subtraction to the ToF. Finally, the signal-processing unit, also known as a microprocessor (MP), receives the ToF from the TDC and computes the distance between the LiDAR sensor and the target object [26].

**Figure 2 sensors-25-02757-f002:**
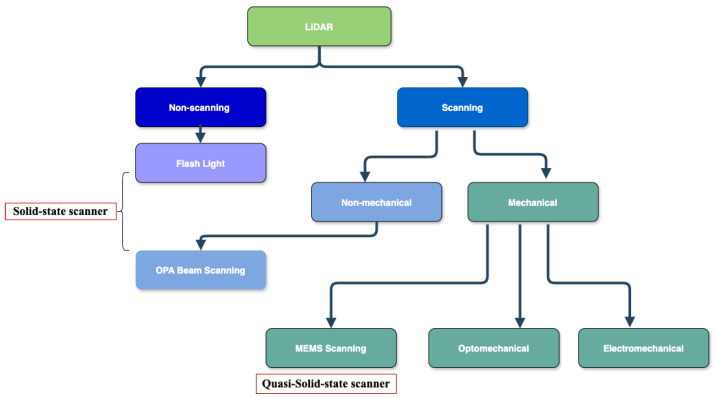
Types of LiDAR sensors. Solid-state LiDAR refers to both non-scanning and non-mechanical scanning LiDAR systems. OPA and flash LiDAR are both solid-state LiDARs since their beam-steering and scanning mechanisms do not have any moving mechanical components. MEMS is known as a quasi-solid-state scanner because, unlike typical LiDAR systems, it lacks massive mechanical moving parts but contains microscopic moving components.

**Table 1 sensors-25-02757-t001:** Comparative analysis of UAV detection technologies.

Sensor Type	Strengths	Limitations	Best Use Cases
Radar [4,5,6]	Long-range detection; differentiates UAVs using micro-Doppler signatures; resistant to adverse weather conditions	Low RCS, high cost, deployment complexity	Airport/military zone monitoring; inclement weather operations
RF [7,8,9,10]	Detects control signals; long-range detection; passive	Depends on active transmissions; susceptible to encryption and jamming; unable to detect fully autonomous UAVs	Security and surveillance applications
Acoustic [11,12,13,14]	Low-cost; energy-efficient solution for LoS-limited environments	Limited performance and detection range due to wind and background noise	Indoor, rural, low-altitude, and LoS-restricted environments
Visual cameras [15,16,17,18,19,20,21,22,23]	Rich visual detail; low-cost and flexible	Sensitivity to lighting, weather, and line-of-sight conditions; lack of depth	Daytime detection; UAV classification
LiDAR [24,25,26,27,28,29,30,31,32,33,34,35,36,37,38,39,40,41,42]	High-resolution 3D spatial data; 3D point cloud generation	Irregular, sparse, high-dimensional data; complicated data processing; computationally intensive	Airspace security; surveillance; autonomous navigation; GNSS-denied zones

**Table 2 sensors-25-02757-t002:** Comparative analysis of LiDAR sensors based on scanning mechanism.

LiDAR Sensor Type	Description	Field of View (FoV)	Scanning Mechanism	Use Cases	Advantages	Limitations
MEMS LiDAR [30,31,32,33,34]	Uses moving micro-mirror plates to steer laser beam in free space while the rest of the system’s components remain motionless	Moderate, depends on mirror steering angle	Quasi-solid-state scanning (a combination of solid-state LiDAR and mechanical scanning )	Autonomous vehicles; drones and robotics; medical imaging; space exploration; mobile devices	Accurate steering with minimal moving components; superior in terms of size, resolution, scanning speed, and cost	Limited range and FoV; sensitivity to vibrations and environmental factors
Optomechanical LiDAR [35,36,37]	Uses mechanical/moving components (mirrors, prisms, or entire sensor heads) to steer the laser beam and scan the environment	Wide FoV (up to 360°)	Rotating and oscillating mirror, spinning prism	Remote sensing, self-driving cars, aerial surveying and mapping, robotics, security	Long range, high accuracy and resolution, wide FoV, fast scanning	Bulky and heavy, high cost and power consumption
Electromechanical LiDAR [38]	Uses electrically controlled motors or actuators to move mechanical parts that allow to steer the laser beam in various directions	Wide FoV (up to 360°)	Mirror, prism, or entire sensor head	Autonomous vehicles, remote sensing, atmospheric studies, surveying, and mapping	Enhanced scanning patterns, wide FoV, moderate cost, long range, high precision, and accuracy	High power consumption, limited durability, bulky
OPA LiDAR [39,40]	Employs optical phased arrays (OPAs) to steer the laser beam without any moving components.	Flexible FoV, electronically controlled, can be narrow or wide	Solid-state beam (non-mechanical) scanning mechanism	Autonomous vehicles; high-precision sensing; compact 3D mapping systems	No moving parts; rapid beam steering; compact size; energy efficiency	Limited steering range and beam quality; high manufacturing costs
Flash LiDAR [41,42]	Employs a broad laser beam and a huge photodetector array to gather 3D data in a single shot	Wide FoV (up to 120° horizontally, 90° vertically)	No scanning mechanism	terrestrial and space applications; advanced driving assistance systems (ADAS);	No moving parts; instantaneous capture; real-time 3D imaging	Limited range; lower resolution; sensitive to light and weather conditions

**Table 3 sensors-25-02757-t003:** Real-world LiDAR applications using deep learning.

Application	Sensors	DL Method	Evaluation Metrics	Environment
Drone self-noise filtering and denoising [60]	airborne LiDAR	PointPillars + synthetic data	percentage and number of removed points	a coastal railway bridge inspection (Ireland)
human localization and human activity recognition (HAR) [61]	2D LiDAR	Conv-LSTM deep network	Accuracy, Precision, Recall, F1 score	Simulated indoor household (Unity + ROS2)
UAV detection and 3D tracking (BVLOS) [62]	LiDAR + RGB Camera	YOLOv2 + Kalman Filter (DL-based fusion)	detection rate and accuracy	Simulated UAV flight (Gazebo)
drone detection, UAV type classification, and 2D/3D trajectory estimation [63]	Stereo fisheye camera, Conic and Peripheral 3D LiDARs, 77 GHz mmWave radar	YOLOv9 + multimodal 3D pose estimation	Mean Square Error (Pose MSE Loss); UAV type calssification accuracy	Real-world outdoor UAV flights (MMUAD challenge)
Indoor UAV tracking [64]	Ouster OS0-128 (LiDAR + signal image)	Signal image fusion + DL model	Average Pose Error (APE); Root Mean Squared Error (RMSE); detection range; frame rate (FPS)	Indoor GNSS-denied area
Rock discontinuity detection (geological engineering) [65]	Terrestrial LiDAR scanner	GoogLeNet CNN + PCA	Accuracy; average dip direction difference and a dip angle difference	Tianjin Longtangou Pumped Storage Power Station (mountainous region in Jizhou District, Tianjin, China)
3D object detection from raw point clouds [66]	RGB-D sensors	hierarchical graph network (HGNet)	mean average precision (mAP) and coefficient of variation for AP (cvAP)	Indoor 3D environments: SUN RGB-D, ScanNet-V2 datasets
Autonomous driving [67]	3D LiDAR	GNN based on neighbor feature alignment mechanism	3D object detection and location performance in terms of three detection difficulty cases (easy, moderate, hard)	Outdoor road scenes from the KITTI benchmark

**Table 4 sensors-25-02757-t004:** Deep learning approaches for LiDAR data processing.

DL Approach	Data Representation	Main Techniques	Strengths	Limitations	Examples
Point-based [68,69]	Point clouds	Directly processes point clouds and captures features using shared MLPs	Direct processing of raw point clouds; efficient for sparse data; prevents voxelization and quantization concerns	Computationally expensive due to large-scale and irregular point clouds	PointNet, PointNet++
Voxel-based [70,71]	Voxel grids	Converts the sparse and irregular point cloud into a volumetric 3D grid of voxels	well-structured representation; easy-to-use 3D CNNs; suitable for capturing global context	High memory usage and computational cost due to voxelization; loss of precision in 3D space due to quantization; loss of detail in sparse data region.	VoxelNet, SECOND
Projection-based [72,73]	Plane (image), spherical, cylindrical, BEV projection	Projects the 3D point cloud onto a 2D plane	Efficient processing using 2D CNNs	Loss of spatial features due to 3D-to-2D projection	RANGENet++, BEV, PIXOR, SqueezeSeg
Graph-based [66,67]	Adjacency matrix, feature matrices, graph Laplacian	Models point clouds as a graph, where each point is regarded as a node and edges represent the interactions between them	Effective for dealing with sparse, non-uniform point clouds; enables both local and global context-aware detection; ideal for capturing spatial relationships between points	High computational complexity due to large point clouds	GNN, HGNet
Hybrid appoach [74,75]	Combination of raw point clouds, voxels, projections, etc.	Combines several methods to improve the accuracy of 3D object detection, segmentation, and classification tasks	Improved object localization and segmentation accuracy; flexibility	High memory and computational resources, complex architecture	PointPillars, MVF

**Table 5 sensors-25-02757-t005:** Challenges in sensor fusion-based UAV detection.

Challenge	Description	Affected Component (s)
UAV shape/material variability [54]	UAVs differ in reflectivity, size, and dynamics.	LiDAR, radar
Sensor calibration and alignment [62,76,80]	Misalignment between sensor outputs (e.g., spatial or temporal offsets) degrades fusion accuracy.	LiDAR, Camera, IMU
Sparse or noisy point clouds [64,76]	Fast UAV motion or occlusions result in low-density, disordered data.	LiDAR
Environmental vulnerability [62,64]	Rain, fog, or low-light conditions degrade sensor reliability.	Camera, LiDAR
Data synchronization [63]	Sensor streams operate at different frame rates, causing latency.	All sensors
High computational load [63,80]	Fusion and DL inference increase latency and resource demand.	Fusion module, DL model
Limited datasets [79]	Few public datasets include synchronized multi-sensor UAV data.	Model training
Sensor failure/dropout [80]	Temporary loss of sensors disrupts detection.	Any sensor node

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
