# Peer review of "LiDAR Technology for UAV Detection: From Fundamentals and Operational Principles to Advanced Detection and Classification Techniques"

_sensors, 2025, doi:10.3390/s25092757_

Round 1

Reviewer 1 Report

Comments and Suggestions for Authors

The manuscript provides a comprehensive and well-structured review of LiDAR technology's role in UAV detection. It covers the theoretical principles behind LiDAR, various scanning mechanisms, and the integration of deep learning techniques for point cloud processing. Additionally, it highlights the importance of multi-sensor fusion for improving UAV detection, particularly in GNSS-denied environments.

Major Concerns:

  1. The manuscript would greatly benefit from a more in-depth exploration of the practical challenges associated with deploying LiDAR systems for UAV detection in real-world environments. Issues such as sensor calibration, environmental factors (e.g., weather and lighting), data sparsity, and noise in the point clouds are not sufficiently addressed. A detailed discussion on how these factors impact LiDAR performance would make the paper more relevant for practical applications."
  2. While the manuscript briefly mentions alternative UAV detection technologies such as radar, RF sensors, and cameras, a more thorough comparative analysis is needed. The paper would be enhanced by a deeper examination of the relative strengths and limitations of LiDAR in comparison to these technologies, particularly in varied operational contexts such as urban environments, areas with high interference, or for detecting small, fast-moving UAVs."
  3. The manuscript lacks concrete performance metrics or experimental validation of the proposed deep learning methods and sensor fusion techniques. The inclusion of benchmarks, such as detection accuracy, false positives, processing times, and scalability in real-world environments, would significantly strengthen the paper and provide evidence for the effectiveness of the proposed methods."
  4. It will be better to add several cases when mentioning the different study topics, for example, in the section of LiDAR Data Processing Techniques, it is highly suggested that the a specific case study is provide to show the detailed application of deep learning techniques, such as the rock joint detection, auto driving, etc.

Minor Concerns:

  1. While the paper discusses the fusion of LiDAR with cameras and radar, it would be beneficial to expand the discussion to include other sensor modalities, such as acoustic sensors, thermal imaging, or hyperspectral imaging. These sensors could further enhance UAV detection in specific scenarios, such as low-visibility conditions, and their inclusion would broaden the paper's applicability."
  2. The manuscript focuses primarily on UAV detection, but it would benefit from citing additional studies from related fields, such as geological engineering, where LiDAR and deep learning have been applied to analyze complex point cloud data. Including references to works like 'Automated Identification of Rock Discontinuities from 3D Point Clouds Using a Convolutional Neural Network' would enrich the paper and provide a broader context for the application of LiDAR technology."
  3. Lack of Practical Case Studies: "Incorporating real-world case studies or practical examples of LiDAR-based UAV detection systems would be a valuable addition. These case studies could illustrate how the theoretical methods and technologies presented in the paper are applied in real-world settings, providing further evidence of their effectiveness."
  4. Some sections of the manuscript could benefit from clearer and more concise language. For example, certain sentences are long and complex, which may hinder the reader’s understanding. Breaking these sentences into smaller, more digestible segments would improve clarity. A possible revision could be: 'The paper explores the integration of multi-modal sensor fusion, combining LiDAR with other complementary modalities. This approach improves UAV detection and tracking in complex environments, such as GNSS-denied zones.’
  5. The manuscript frequently uses passive voice, which can make some sections less engaging and harder to follow. Shifting to active voice in key parts of the paper could improve readability and make the writing more direct. For example, 'The data is processed using deep learning techniques to classify UAVs' could be revised to 'We process the data using deep learning techniques to classify UAVs.'
  6. The manuscript uses inconsistent terminology in some instances, particularly when referring to 'LiDAR-based UAV detection' versus 'LiDAR for UAV tracking.' Consistency in terminology is important for clarity, and I suggest ensuring that terms like 'detection,' 'tracking,' and 'localization' are used consistently throughout the paper.
  7. There are several minor grammatical errors, such as missing articles or subject-verb agreement issues. For instance, 'The fusion of multiple sensors have the potential...' should be revised to 'The fusion of multiple sensors has the potential...' While these issues are minor, correcting them would improve the overall quality of the paper.
  8. Inconsistent Use of Capitalization: "I noticed some inconsistencies in capitalization, particularly with terms like 'Deep Learning' versus 'deep learning.' Ensuring that these terms are consistently written will help maintain a professional tone throughout the manuscript.

Comments on the Quality of English Language
  1. Some sections of the manuscript could benefit from clearer and more concise language. For example, certain sentences are long and complex, which may hinder the reader’s understanding. Breaking these sentences into smaller, more digestible segments would improve clarity. A possible revision could be: 'The paper explores the integration of multi-modal sensor fusion, combining LiDAR with other complementary modalities. This approach improves UAV detection and tracking in complex environments, such as GNSS-denied zones.’
  2. The manuscript frequently uses passive voice, which can make some sections less engaging and harder to follow. Shifting to active voice in key parts of the paper could improve readability and make the writing more direct. For example, 'The data is processed using deep learning techniques to classify UAVs' could be revised to 'We process the data using deep learning techniques to classify UAVs.'
  3. The manuscript uses inconsistent terminology in some instances, particularly when referring to 'LiDAR-based UAV detection' versus 'LiDAR for UAV tracking.' Consistency in terminology is important for clarity, and I suggest ensuring that terms like 'detection,' 'tracking,' and 'localization' are used consistently throughout the paper.
  4. There are several minor grammatical errors, such as missing articles or subject-verb agreement issues. For instance, 'The fusion of multiple sensors have the potential...' should be revised to 'The fusion of multiple sensors has the potential...' While these issues are minor, correcting them would improve the overall quality of the paper.
  5. I noticed some inconsistencies in capitalization, particularly with terms like 'Deep Learning' versus 'deep learning.' Ensuring that these terms are consistently written will help maintain a professional tone throughout the manuscript.

Author Response

Dear Reviewer,

We sincerely appreciate your time and effort in reviewing our manuscript. Your comments were insightful and constructive, and they have helped us significantly improve the quality, clarity, and practical relevance of the paper. Below we provide a detailed, point-by-point response to each of your major and minor concerns.

Major Concerns:

  1. The manuscript would greatly benefit from a more in-depth exploration of the practical challenges associated with deploying LiDAR systems for UAV detection in real-world environments. Issues such as sensor calibration, environmental factors (e.g., weather and lighting), data sparsity, and noise in the point clouds are not sufficiently addressed. A detailed discussion on how these factors impact LiDAR performance would make the paper more relevant for practical applications."

Response: Thank you for this constructive recommendation. In response, we have included a dedicated subsection titled 3.4.2. Challenges in Sensor Fusion-Based UAV Detection, where we outline key challenges encountered in real-world deployments.

  1. While the manuscript briefly mentions alternative UAV detection technologies such as radar, RF sensors, and cameras, a more thorough comparative analysis is needed. The paper would be enhanced by a deeper examination of the relative strengths and limitations of LiDAR in comparison to these technologies, particularly in varied operational contexts such as urban environments, areas with high interference, or for detecting small, fast-moving UAVs."

Response: Thank you for your valuable comment. While the manuscript already discussed the main types of UAV detection sensors, we agree that a clearer comparative overview would enhance the clarity and practical relevance of the paper. In response, we have added a comparative summary table that outlines the advantages, disadvantages, and best use cases of LiDAR, radar, RF sensors, acoustic sensors, and cameras. This table provides a more structured and accessible comparison of these technologies, especially in relation to urban environments, high-interference areas, and the detection of small, fast-moving UAVs, as you suggested. A detailed comparison is provided in Table 1, which outlines the strengths, limitations, and use cases of various UAV detection technologies (Please, see “Table 1. Comparative Analysis of UAV Detection Technologies”, for more information). 

  1. The manuscript lacks concrete performance metrics or experimental validation of the proposed deep learning methods and sensor fusion techniques. The inclusion of benchmarks, such as detection accuracy, false positives, processing times, and scalability in real-world environments, would significantly strengthen the paper and provide evidence for the effectiveness of the proposed methods."

Response: We thank the reviewer for this valuable suggestion. In response, we have revised the manuscript by introducing a new Subsection 3.3 and adding Table 3: Real-world LiDAR Applications Using Deep Learning, which presents different performance metrics extracted from the analyzed studies. These metrics include Mean Square Error (Pose MSE Loss), UAV classification accuracy, Average Pose Error (APE), Root Mean Squared Error (RMSE), detection range, and frame rate (FPS), among others. For further details, please refer to Table 3 and Subsection 3.3.

  1. It will be better to add several cases when mentioning the different study topics, for example, in the section of LiDAR Data Processing Techniques, it is highly suggested that the a specific case study is provide to show the detailed application of deep learning techniques, such as the rock joint detection, auto driving, etc.

 Response: Thank you for your helpful suggestion. In response, we have expanded the LiDAR Data Processing Techniques section by incorporating several practical case studies that illustrate how deep learning methods are applied to real-world scenarios. These examples aim to demonstrate the versatility and effectiveness of LiDAR-based learning techniques across domains. For more details, please refer to Subsection 3.3 and Table 3.

Minor Concerns:

  1. While the paper discusses the fusion of LiDAR with cameras and radar, it would be beneficial to expand the discussion to include other sensor modalities, such as acoustic sensors, thermal imaging, or hyperspectral imaging. These sensors could further enhance UAV detection in specific scenarios, such as low-visibility conditions, and their inclusion would broaden the paper's applicability."

Response: Thank you for your thoughtful suggestion. Our manuscript already includes a detailed discussion on the use of acoustic sensors and their integration with deep learning models for UAV detection. While we had not previously addressed the fusion of LiDAR and acoustic sensors, we have now expanded the discussion to include thermal imaging as an additional modality that can improve detection in low-visibility scenarios. These updates help broaden the scope of the paper and outline future directions for multi-sensor UAV detection frameworks:

“Recent advances in multi-sensor fusion have shown that the integration of several sensor modalities, such as LiDAR, infrared (IR), visible cameras, and acoustic sensors, may considerably improve the robustness and accuracy of UAV detection systems [84]. Each sensor provides complementary data: LiDAR provides high-resolution spatial and depth information, infrared sensors capture thermal signatures useful in low-light or obscured conditions, video cameras provide rich visual context, and acoustic sensors detect UAV-specific sound patterns, particularly in scenarios where visual line-of-sight is limited”.

Although fusion of LiDAR and acoustic sensors is still a developing field, recent research such as Semenyuk et al. [84] see this integration as a potential path for enhancing UAV detection robustness in noisy and GNSS-denied environments. They emphasize the possibility of merging audio characteristics with LiDAR-based spatial data, employing sophisticated sensor fusion and deep learning techniques.

  1. The manuscript focuses primarily on UAV detection, but it would benefit from citing additional studies from related fields, such as geological engineering, where LiDAR and deep learning have been applied to analyze complex point cloud data. Including references to works like 'Automated Identification of Rock Discontinuities from 3D Point Clouds Using a Convolutional Neural Network' would enrich the paper and provide a broader context for the application of LiDAR technology."

Response: Thank you for your helpful suggestion. In response, we have added a reference to the work by Ge et al. (2025) in the Introduction section to highlight the broader applicability of LiDAR and CNN-based deep learning methods beyond UAV detection. This study demonstrates how similar techniques have been successfully used in geological engineering for the automated identification of rock discontinuities from LiDAR-generated point clouds. Including this reference helps to provide a broader context for the application of LiDAR and deep learning technologies in diverse domains.

Beyond UAV detection, LiDAR and the GoogLeNet CNN-based frameworks have been effectively used in other fields, such as geological engineering. Ge et al. [82] accurately identified rock joints in 3D point clouds and determined their orientation. The proposed model obtained high classification accuracy while simply utilizing point coordinates and normals as input, and it was validated on two real-world datasets. The trained model identified three natural discontinuity groups, and individual discontinuities were extracted using the density-based spatial clustering of applications with noise (DBSCAN) algorithm. Their orientations were then determined using principal component analysis (PCA) algorithm”.

  1. Lack of Practical Case Studies: "Incorporating real-world case studies or practical examples of LiDAR-based UAV detection systems would be a valuable addition. These case studies could illustrate how the theoretical methods and technologies presented in the paper are applied in real-world settings, providing further evidence of their effectiveness."

Response: We thank the reviewer for this valuable comment. In response, we have significantly revised the manuscript to address the need for real-world case studies that demonstrate the practical implementation of LiDAR-based UAV detection systems.

A new subsection titled 3.3 “Real-World Applications of LiDAR with Deep Learning” has been added, summarizing eight representative case studies from recent literature. These include applications such as drone swarm detection using PointPillars [68], UAV localization in GNSS-denied environments using YOLOv2 with Kalman fusion [70], and multimodal 3D pose estimation with YOLOv9 and radar-LiDAR-camera fusion [73]. Additional cases involve indoor UAV tracking [74], motion recognition using Conv-LSTM [69], rock joint detection with GoogLeNet CNN [82], and urban scene classification using DGCNN and ConvPoint [58]. High-performing architectures for autonomous driving, such as PointNet++, VoxelNet, and OctNet, are also included [59].

Each case highlights how the discussed deep learning models have been successfully applied in operational settings, often under challenging conditions such as GNSS-denied environments, complex urban scenes, and unstructured terrains. These case studies are now summarized in a clearly formatted table (Table 3) and discussed in detail within the revised text. We believe these additions strengthen the manuscript by bridging theoretical analysis with real-world deployment, as recommended.

  1. Some sections of the manuscript could benefit from clearer and more concise language. For example, certain sentences are long and complex, which may hinder the reader’s understanding. Breaking these sentences into smaller, more digestible segments would improve clarity. A possible revision could be: 'The paper explores the integration of multi-modal sensor fusion, combining LiDAR with other complementary modalities. This approach improves UAV detection and tracking in complex environments, such as GNSS-denied zones.’

Response: Thank you for your valuable suggestion. We have revised the last long sentence in the Abstract to enhance clarity and readability by breaking it into two shorter, more digestible parts. The updated version reads:

“The paper explores the integration of multi-modal sensor fusion, combining LiDAR with other complementary modalities. This approach improves UAV detection and tracking in complex environments, such as global navigation satellite system (GNSS)-denied zones”.

  1. The manuscript frequently uses passive voice, which can make some sections less engaging and harder to follow. Shifting to active voice in key parts of the paper could improve readability and make the writing more direct. For example, 'The data is processed using deep learning techniques to classify UAVs' could be revised to 'We process the data using deep learning techniques to classify UAVs.'

Response: We appreciate the reviewer’s suggestion regarding the use of active voice. While the exact sentence mentioned was not present in our manuscript, we acknowledge that several sections could benefit from a more direct and engaging tone. We have revised multiple instances of passive voice throughout the paper by shifting to active constructions where appropriate. These changes enhance readability and clarity while maintaining the academic tone of the manuscript.

Line number

Initial version

Revised version

73 (previous)

Deep learning is being used to recognize drone acoustic signatures, which is an important area of object detection development.

The use of deep learning to recognize drone sound signatures represents a major improvement in object detection.

363 (previous)

To test the proposed algorithm’s efficacy a simulation study using the Gazebo simulator was carried out by performing a thorough comparison between the proposed method and state-of-the-art techniques such as FUEL (Fast UAV Exploration) and FAEP (Fast Autonomous Exploration Planner).

The authors evaluated the suggested algorithm's efficacy using the Gazebo simulator, comparing it to cutting-edge solutions such as FUEL (Fast UAV Exploration) and FAEP (Fast Autonomous Exploration Planner).

382 (previous)

Efficient moving object detection based on 2D LiDAR combined with frame-to frame scan matching method was presented in [48].

Reference [48] presents an effective moving object recognition approach based on 2D LiDAR and frame-to-frame scan matching.

390 (previous)

To detect dynamic objects, the proposed algorithm was tested on a stationary measurement, during performing SLAM and in real-time measurements respectively.

The authors evaluated the proposed approach for dynamic object detection in three scenarios: stationary measurements, SLAM execution, and real-time measurements.

384 (previous)

To ensure collision-free passage across a mapped area, the proposed SegMatch algorithm was implemented on autonomous mobile robotic system (MRS) equipped with LD-OEM 1000 LiDAR.

The researchers used an autonomous mobile robotic system (MRS) with an LD-OEM 1000 LiDAR to test the suggested SegMatch algorithm. This configuration was used to assure collision-free passage across a mapped region.

  1. The manuscript uses inconsistent terminology in some instances, particularly when referring to 'LiDAR-based UAV detection' versus 'LiDAR for UAV tracking.' Consistency in terminology is important for clarity, and I suggest ensuring that terms like 'detection,' 'tracking,' and 'localization' are used consistently throughout the paper.

Response: Thank you for your helpful feedback. We have reviewed the manuscript and revised the terminology to ensure consistent and accurate use of terms such as detection, tracking, and localization throughout the paper.

Line 105

In addition, LiDAR sensors have shown great potential for detecting, localizing, and tracking objects in the near- and mid-range.

In addition, LiDAR sensors have shown promising results for UAV detection, localization, and tracking in both near- and mid-range scenarios.

Line 421

Additionally, the potential of 3D Lidar for UAV tracking depending on lighting conditions was estimated, respectively.

Additionally, the potential of 3D LiDAR for UAV detection and tracking under lighting conditions was evaluated.

  1. There are several minor grammatical errors, such as missing articles or subject-verb agreement issues. For instance, 'The fusion of multiple sensors have the potential...' should be revised to 'The fusion of multiple sensors has the potential...' While these issues are minor, correcting them would improve the overall quality of the paper.

Response: Thank you for your careful review and for pointing out potential grammatical issues. We have thoroughly reviewed the manuscript for subject-verb agreement and article usage. While the specific example ‘The fusion of multiple sensors have the potential...’ does not appear in the manuscript, we appreciate your attention to detail. We have nonetheless carefully revised the manuscript to correct minor grammatical errors and improve overall clarity and readability.

  1. Inconsistent Use of Capitalization: "I noticed some inconsistencies in capitalization, particularly with terms like 'Deep Learning' versus 'deep learning.' Ensuring that these terms are consistently written will help maintain a professional tone throughout the manuscript.

Response: Thank you for your valuable observation. We have carefully reviewed the manuscript and corrected all inconsistencies in capitalization. Terms such as "deep learning" are now used consistently in lowercase throughout the text, in accordance with standard academic writing conventions. These revisions help enhance the professionalism and clarity of the manuscript.

Comments on the Quality of English Language

  1. Some sections of the manuscript could benefit from clearer and more concise language. For example, certain sentences are long and complex, which may hinder the reader’s understanding. Breaking these sentences into smaller, more digestible segments would improve clarity. A possible revision could be: 'The paper explores the integration of multi-modal sensor fusion, combining LiDAR with other complementary modalities. This approach improves UAV detection and tracking in complex environments, such as GNSS-denied zones.’

Response: Thank you for your valuable suggestion. We have revised the last long sentence in the Abstract to enhance clarity and readability by breaking it into two shorter, more digestible parts. The updated version reads:

“The paper explores the integration of multi-modal sensor fusion, combining LiDAR with other complementary modalities. This approach improves UAV detection and tracking in complex environments, such as global navigation satellite system (GNSS)-denied zones”.

  1. The manuscript frequently uses passive voice, which can make some sections less engaging and harder to follow. Shifting to active voice in key parts of the paper could improve readability and make the writing more direct. For example, 'The data is processed using deep learning techniques to classify UAVs' could be revised to 'We process the data using deep learning techniques to classify UAVs.'

Response: We appreciate the reviewer’s suggestion regarding the use of active voice. While the exact sentence mentioned was not present in our manuscript, we acknowledge that several sections could benefit from a more direct and engaging tone. We have revised multiple instances of passive voice throughout the paper by shifting to active constructions where appropriate. These changes enhance readability and clarity while maintaining the academic tone of the manuscript.

Line number

Initial version

Revised version

73 (previous)

Deep learning is being used to recognize drone acoustic signatures, which is an important area of object detection development.

The use of deep learning to recognize drone sound signatures represents a major improvement in object detection.

363 (previous)

To test the proposed algorithm’s efficacy a simulation study using the Gazebo simulator was carried out by performing a thorough comparison between the proposed method and state-of-the-art techniques such as FUEL (Fast UAV Exploration) and FAEP (Fast Autonomous Exploration Planner).

The authors evaluated the suggested algorithm's efficacy using the Gazebo simulator, comparing it to cutting-edge solutions such as FUEL (Fast UAV Exploration) and FAEP (Fast Autonomous Exploration Planner).

382 (previous)

Efficient moving object detection based on 2D LiDAR combined with frame-to frame scan matching method was presented in [48].

Reference [48] presents an effective moving object recognition approach based on 2D LiDAR and frame-to-frame scan matching.

390 (previous)

To detect dynamic objects, the proposed algorithm was tested on a stationary measurement, during performing SLAM and in real-time measurements respectively.

The authors evaluated the proposed approach for dynamic object detection in three scenarios: stationary measurements, SLAM execution, and real-time measurements.

384 (previous)

To ensure collision-free passage across a mapped area, the proposed SegMatch algorithm was implemented on autonomous mobile robotic system (MRS) equipped with LD-OEM 1000 LiDAR.

The researchers used an autonomous mobile robotic system (MRS) with an LD-OEM 1000 LiDAR to test the suggested SegMatch algorithm. This configuration was used to assure collision-free passage across a mapped region.

  1. The manuscript uses inconsistent terminology in some instances, particularly when referring to 'LiDAR-based UAV detection' versus 'LiDAR for UAV tracking.' Consistency in terminology is important for clarity, and I suggest ensuring that terms like 'detection,' 'tracking,' and 'localization' are used consistently throughout the paper.

Response: Thank you for your helpful feedback. We have reviewed the manuscript and revised the terminology to ensure consistent and accurate use of terms such as detection, tracking, and localization throughout the paper.

Line 105

In addition, LiDAR sensors have shown great potential for detecting, localizing, and tracking objects in the near- and mid-range.

In addition, LiDAR sensors have shown promising results for UAV detection, localization, and tracking in both near- and mid-range scenarios.

Line 421

Additionally, the potential of 3D Lidar for UAV tracking depending on lighting conditions was estimated, respectively.

Additionally, the potential of 3D LiDAR for UAV detection and tracking under lighting conditions was evaluated.

  1. There are several minor grammatical errors, such as missing articles or subject-verb agreement issues. For instance, 'The fusion of multiple sensors have the potential...' should be revised to 'The fusion of multiple sensors has the potential...' While these issues are minor, correcting them would improve the overall quality of the paper.

Response: Thank you for your careful review and for pointing out potential grammatical issues. We have thoroughly reviewed the manuscript for subject-verb agreement and article usage. While the specific example ‘The fusion of multiple sensors have the potential...’ does not appear in the manuscript, we appreciate your attention to detail. We have nonetheless carefully revised the manuscript to correct minor grammatical errors and improve overall clarity and readability.

  1. I noticed some inconsistencies in capitalization, particularly with terms like 'Deep Learning' versus 'deep learning.' Ensuring that these terms are consistently written will help maintain a professional tone throughout the manuscript.

Response: Thank you for your valuable observation. We have carefully reviewed the manuscript and corrected all inconsistencies in capitalization. Terms such as "deep learning" are now used consistently in lowercase throughout the text, in accordance with standard academic writing conventions. These revisions help enhance the professionalism and clarity of the manuscript.

p/s: We sincerely thank all reviewers for their thoughtful feedback and constructive suggestions. In response, we have made substantial improvements to the manuscript, including the following key changes:

  • The overall structure of the paper has been revised for improved clarity and logical flow. Several sections were reorganized and merged to avoid redundancy and enhance coherence.
  • New tables have been added to summarize real-world applications, performance metrics, and comparative analysis of deep learning-based UAV detection methods.
  • Additional literature sources have been reviewed and incorporated to strengthen the discussion on practical challenges, LiDAR limitations, and sensor fusion strategies.
  • Grammatical and stylistic corrections were made throughout the manuscript to ensure a more polished and professional presentation.

We hope these comprehensive revisions have significantly improved the quality and readability of the manuscript, and we hope that the updated version will meet the expectations and standards of the journal.

With sincere appreciation,

The Authors

Reviewer 2 Report

Comments and Suggestions for Authors

Very interesting review! I only have some minor suggestions to improve the overall quality of the manuscript

Comments on the Quality of English Language

It can be improved in some details

Author Response

Dear Reviewer,

Thank you very much for taking the time to provide detailed and thoughtful feedback on our manuscript titled “LiDAR Technology for UAV Detection: From Fundamentals and Operational Principles to Advanced Detection and Classification Techniques.” We appreciate your insights, which have helped us improve the clarity, precision, and consistency of the manuscript.

Please find below our point-by-point response and a summary of the corresponding revisions made.

General comments

  1. “The description is a bit qualitative in some stages. The use of expressions like ‘tiny objects’, ‘huge data’, or ‘weak reflected light pulse’ should always be accompanied by a precise quantitative estimate of the physical dimension.”

Response: We appreciate the reviewer’s observation regarding the use of qualitative language. In response, we have revised the manuscript to include more precise and quantitative descriptions, as follows:

  • “Tiny objects” has been clarified as small-scale objects like pedestrians, drones, small animals, which are more difficult to detect due to limited reflectivity and surface area.

Lines 199-201: It also boosts low-intensity return signals from small-scale objects like pedestrians, drones, small animals, which is essential in long-range LiDAR systems or low-reflectivity environments where point cloud density is low.

  • The phrase ‘weak reflected light pulse’ has been clarified to refer to low-intensity reflected light signals, typically requiring photon-level sensitivity, such as those detected by SPADs (Single-Photon Avalanche Diodes) in long-range or low-reflectivity LiDAR applications.

Lines 191-194: Avalanche photodiodes (APDs) and single-photon avalanche diodes (SPADs) are commonly used detectors that capture low-intensity reflected light signals—often with photon-level sensitivity – and convert them into electrical signals for further processing

  • We have revised the manuscript to replace the qualitative expression “huge data” with a more precise and measurable description. The revised sentence now reads:

Lines 205-206: “In some cases, the signal processing unit performs data compression and transmission tasks to manage large-scale LiDAR data – often exceeding 100 GB—in real-time applications.”

  1. “The acronym ‘LiDAR’ is written differently throughout the manuscript (LiDAR, Lidar, lidar). Please give thorough proof of the manuscript and correct these details.”

Response: Thank you for pointing this out. We have thoroughly reviewed the manuscript and standardized the usage of the acronym to “LiDAR” throughout the text to ensure consistency and maintain scientific formatting.

Line-by-Line Revisions

  • Line 3: Please either use “due to” or “because of” instead of “because to”.

Response: Thank you for catching this grammatical issue. We have revised the phrase by replacing “because to” with “due to” to ensure correct usage and improve sentence clarity.

  • Lines 11-12: Report the acronym GNSS.

Response: Thank you for your suggestion. The acronym GNSS has been expanded to Global Navigation Satellite System upon its first appearance in the manuscript to ensure clarity for all readers.

  • Lines 77–79: Report the acronyms DNN, CNN and RNN.

Response: Thank you for the suggestion. We have expanded all acronyms upon their first appearance in the manuscript. Specifically, DNN is now written as Deep Neural Network, CNN as Convolutional Neural Network, and RNN as Recurrent Neural Network. This clarification has been made to enhance readability for a broader audience.

4) Line 102: The acronym LiDAR means Light Detection and Ranging. Please correct.

Response: Thank you for pointing this out. We have corrected the definition of the acronym “LiDAR” to its accurate form — Light Detection and Ranging — at its first mention in the manuscript to ensure clarity and correctness.

5) Line 124: For the sake of clarity, I would add a short paragraph at the end of the Introduction describing the content of each of the remaining Sections.

Response: We appreciate the reviewer’s helpful suggestion. In response, we have added a summary paragraph at the end of the Introduction outlining the content of each major section in the manuscript to improve clarity and assist readers in navigating the structure of the paper.

6) Lines 131-133: The sentence “how long it takes allowing the reflected light […]” is a bit unclear. Please rephrase it as: “measuring the time necessary to the light beam to reach the target” or something similar.

Response: Thank you for this helpful suggestion. We have revised the sentence to improve clarity. It now reads:

“LiDAR systems can generate detailed 3D models of the environment by emitting laser pulses and measuring the time required for the light beam to reach the target and reflect back.” This phrasing follows your recommendation and enhances both grammatical correctness and scientific clarity.

7) Lines 172–173: Please remove this sentence as it sounds redundant (it has already been stated that CW LiDARs send a continuous light wave).

Response: Thank you for pointing this out. We have carefully revised the entire paragraph discussing pulsed and continuous-wave (CW) LiDAR systems to improve clarity and avoid redundancy. The sentence stating that CW lasers produce a continuous beam was removed, as this had already been introduced earlier in the section. In addition, we enhanced the overall structure and language of the paragraph to provide a clearer explanation of the operational principles of AMCW and FMCW LiDAR, including distinctions in their accuracy and range limitations. These updates also ensure a smoother flow between concepts and more consistent terminology.

8) Line 191: How does this distinction work? Is it based on the signal to noise ratio? Please explain.

Response: Thank you for your helpful comment. We have clarified that the distinction refers to low intensity return signals, which are typical when LiDAR scans small-scale objects (e.g., drones, pedestrians, small animals) or distant targets. These signals often have a low signal-to-noise ratio (SNR) due to weak reflectivity or long travel distance, making them harder to detect reliably. The signal processing unit plays a key role in boosting such low-SNR signals, enabling accurate detection and 3D mapping even when point cloud density is low. This is particularly important in long-range LiDAR systems or low-reflectivity environments, as also noted in [79, 80, 81]. We revised the sentence in the manuscript accordingly to clarify this technical distinction.

9) Line 222: Please state the difference between scanning and non-scanning LiDARs.

Response (Lines 231-235): We have clarified the distinction between scanning and non-scanning LiDARs as follows: Scanning LiDARS use either mechanical or non-mechanical systems to sequentially steer the laser beam and scan the surrounding environment. In contrast, non-scanning LiDAR systems do not rely on any scanning mechanisms to move the laser beam across its field of view (FoV). Instead, they illuminate the full scene in a single light pulse and capture its reflection on a 2D sensor array, similar to a camera [30].

10) Line 504-505: The expression “in the direction of the drone” is unclear. Do you mean “with respect to the drone” or “relative to the drone”?

Response (Lines 515-516): Thank you for this observation. We agree that the original phrasing was ambiguous. The sentence has been revised for clarity and now uses the expression “relative to the drone’s changing position”, which more accurately conveys the intended meaning. The updated sentence: “Because the LiDAR sensor is mounted on a moving drone, it continuously scans the environment relative to the drone’s changing position and orientation”.

11) Line 532: By (x, y, z) do you mean Cartesian coordinates? If so, please refer to it as “Cartesian coordinates”.

Response: Thank you for the clarification request. We have revised the sentence to explicitly refer to the (x, y, z) coordinates as Cartesian coordinates, as suggested. The updated sentence: “The UAV’s position is estimated using the mean of the selected point cloud cluster, expressed in Cartesian coordinates (x, y, z)”.

p/s: We sincerely thank all reviewers for their thoughtful feedback and constructive suggestions. In response, we have made substantial improvements to the manuscript, including the following key changes:

  • The overall structure of the paper has been revised for improved clarity and logical flow. Several sections were reorganized and merged to avoid redundancy and enhance coherence.
  • New tables have been added to summarize real-world applications, performance metrics, and comparative analysis of deep learning-based UAV detection methods.
  • Additional literature sources have been reviewed and incorporated to strengthen the discussion on practical challenges, LiDAR limitations, and sensor fusion strategies.
  • Grammatical and stylistic corrections were made throughout the manuscript to ensure a more polished and professional presentation.

We hope these comprehensive revisions have significantly improved the quality and readability of the manuscript, and we hope that the updated version will meet the expectations and standards of the journal.

Sincerely,
Dr. Ulzhalgas Seidaliyeva
(on behalf of all authors)

Reviewer 3 Report

Comments and Suggestions for Authors

The authors systematically reviewed the recent innovations in LiDAR-based drone detection. At the first look, it seems a good conclusion for the LiDAR-based UAV detection development. But after reading carefully, there are some major flaws that need to be addressed. I recommend a major revision.

  1. The authors want to emphasize the the recent innovations in LiDAR-based drone detection. Thus, they should concentrate on the UAV features introduction, how to identify them, and precision evaluation, with a combination of some figures, not directly and only give some dry writing.
  2. Besides, the figures has a very low resolution, and the quality are not high enough for a scientific paper.
  3. For Line 21, , "in the first quarter of this year", in which year, the authors should get rid of such vague expression.
  4. For the reference [1], now it cannot be accessed. Please confirm such news report should not be a right reference for a paper.
  5. The title of Section 5 is inappropriate. Discussion is discussion, while Conclusions are only conclusion. I cannot position the final conclusion point for such a paper. Besides, I don' t get your real and deep understanding for the research prospect in this field.

Author Response

We sincerely thank the reviewer for the time and thoughtful feedback. We appreciate your detailed comments, which helped us improve the quality, clarity, and scientific value of our manuscript. Below we provide point-by-point responses and explain how we revised the paper accordingly.

  1. The authors want to emphasize the recent innovations in LiDAR-based drone detection. Thus, they should concentrate on the UAV feature’s introduction, how to identify them, and precision evaluation, with a combination of some figures, not directly and only give some dry writing.

Answer: We appreciate the reviewer’s feedback. In response, we have enriched Table 3 to highlight recent innovations in UAV detection using LiDAR, including the specific features used for identification, model types, and corresponding evaluation metrics such as detection accuracy, precision, and recall. This addition provides a clearer, more practical summary of how LiDAR-based systems detect UAVs and assess performance in real-world scenarios.

  1. Besides, the figures have a very low resolution, and the quality are not high enough for a scientific paper.

Answer: We appreciate your observation. We agree and have updated all figures in the revised manuscript. High-resolution versions have been inserted to ensure better visual clarity and consistency with scientific publishing standards. Updated figures appear in Sections 2 through 5.

  1. For Line 21, "in the first quarter of this year", in which year, the authors should get rid of such vague expression.

Answer: Thank you for the comment. The original sentence referred to data from early 2024, but to ensure clarity and relevance, we have updated the text using more recent statistics from the first quarter of 2025. According to FAA reports, more than 400 drone-related incidents occurred near U.S. airports during this period. The text and reference have been revised accordingly. The updated sentence now reads:

For example, in the first quarter of 2025, the FAA (Federal Aviation Administration) reported more than 400 drone-related near-miss incidents near U.S. airports, with multiple cases requiring pilots to take evasive action [1].”

  1. For the reference [1], now it cannot be accessed. Please confirm such news report should not be a right reference for a paper.

Answer: Thank you for pointing this out. We agree with the reviewer that news sources may not be suitable for a scholarly manuscript. Therefore, we have removed Reference [1] and replaced it with updated statistical data from the official FAA incident records for the first quarter of 2025. This improves both the reliability and academic quality of the citation (see updated References section).

  1. The title of Section 5 is inappropriate. Discussion is discussion, while Conclusions are only conclusion. I cannot position the final conclusion point for such a paper. Besides, I don' t get your real and deep understanding for the research prospect in this field.

Answer: We appreciate this comment. Section 5 has been split into two distinct parts: Section 4 – Discussion and Section 5 – Conclusion and Future Directions.  The Discussion section focuses on analysis and interpretation of the reviewed studies, while the Conclusion summarizes the key insights and explicitly highlights the research contributions. These changes improve structural clarity and better communicate our insights and perspective on the field.

p/s: We sincerely thank all reviewers for their thoughtful feedback and constructive suggestions. In response, we have made substantial improvements to the manuscript, including the following key changes:

  • The overall structure of the paper has been revised for improved clarity and logical flow. Several sections were reorganized and merged to avoid redundancy and enhance coherence.
  • New tables have been added to summarize real-world applications, performance metrics, and comparative analysis of deep learning-based UAV detection methods.
  • Additional literature sources have been reviewed and incorporated to strengthen the discussion on practical challenges, LiDAR limitations, and sensor fusion strategies.
  • Grammatical and stylistic corrections were made throughout the manuscript to ensure a more polished and professional presentation.

We hope these comprehensive revisions have significantly improved the quality and readability of the manuscript, and we hope that the updated version will meet the expectations and standards of the journal.

With kind regards,
The Authors

Round 2

Reviewer 1 Report

Comments and Suggestions for Authors

All of my concerns have been addressed by the authors.